# Power Generation Performance of Building-Integrated Photovoltaic Systems in a Zero Energy Building

**Won Jun Choi** [1] , **Hong Jin Joo** [2] , **Jae-Wan Park** [3,*] , **Sang-kyun Kim** [1] **and Jae-Bum Lee** [1]

[1]   National Institute of Environmental Research, Incheon 22689, Korea; choiwj@me.go.kr (W.J.C.); nierkum@korea.kr (S.-k.K.); gercljb@korea.kr (J.-B.L.)
[2]   Solar Thermal Convergence Laboratory, Korea Institute of Energy Research, Daejeon 34129, Korea; joo@kier.re.kr
[3]   Research and Development Office, TES Eng, Daejeon 35245, Korea
*   Correspondence: mil0516@hanmail.net; Tel.: +82-42-280-2599

**Abstract:** In this study, the long-term operational performance of building-integrated photovoltaic (BIPV) systems was analyzed in the Carbon Zero Building of the National Institute of Environmental Research (NIER) of South Korea, with a total area of 2449 m$^2$. Three types of BIPV modules (glass to glass, glass to Tedlar/crystal, and amorphous) were installed in the building envelopes (roofs, walls, windows, atrium, and pergola) with a total capacity of 116.2 kWp. Over a five-year period, the average annual energy production was 855.6 kWh/kWp, the system loss ranged from 0.14 to 0.31 h/d, and the capture loss ranged from 0.21 to 1.81 h/d. The causes of capture losses were degradation of the power generation efficiency of the horizontal installation module due to the accumulation of dust and reduced energy production due to application of the same inverter for the crystal system module and amorphous module. As a result, the BIPV systems with an installation angle of 30° exhibited approximately 57% higher energy production than vertically (90°) installed systems under the same solar radiation. Moreover, horizontal (0°) BIPV systems exhibited up to 14% higher energy production than vertical BIPV systems.

**Keywords:** Building-Integrated Photovoltaic Systems; Renewable Energy; Carbon Zero Building; energy generation; South Korea

---

## 1. Introduction

With growing concerns over global warming due to greenhouse gas emissions from the burning of fossil fuels, countries are showing increasing interest in energy saving measures and the development of eco-friendly energy. According to the report published by the Global Alliance for Buildings and Construction in 2017, the energy used in buildings and building construction accounts for 36% of the final energy consumption and 39% of greenhouse gas emissions [1]. This means that buildings consume the most energy but can also represent the greatest energy savings. Accordingly, interest in zero energy buildings, which can significantly contribute to energy saving and greenhouse gas reduction, is rapidly rising, with many countries supporting building energy saving research, development, and policies led by the government. For example, the European Energy Performance of Building Directive aims to achieve zero energy in all new buildings by the end of 2020, and the DOE (Department of Energy) of the United States aims to construct zero energy houses with economic efficiency by 2020 and achieve zero energy in all new buildings by 2025 [2,3].

To achieve zero energy buildings, it is crucial to establish a strategy according to the energy consumption pattern of the target building. In addition, an energy saving technology suitable for the

building characteristics must be introduced and a method of supplying renewable energy must be presented [4]. In particular, energy saving technology has been developed with a focus on element technology to reduce the cooling and heating energy of a building. For zero energy buildings that first applied element technology, the proportion of electric energy increases as the use of thermal energy decreases [5]. As electricity has emerged as an important energy source for achieving zero energy buildings, renewable energy technology capable of producing and supplying electricity from the building itself is continuously developing. Among the current commercial renewable energy technologies, photovoltaics, solar thermal power generation, bio-pellet cogeneration, fuel cells, and wind power generation are considered applicable to buildings. Among these technologies, systems directly applicable to non-residential buildings are very limited due to issues related to system reliability and safety, load response level, installation situations, and economic efficiency compared to performance [6].

Considering current technologies and market situations, the most promising renewable energy technology for producing electric energy from buildings is photovoltaics, despite its somewhat low efficiency. Among photovoltaic technologies, building-integrated photovoltaic (BIPV) technology, which integrates the functions of building exterior materials and photovoltaic (PV) modules into building envelopes, has been the subject of widespread research and development related to zero energy buildings [7,8].

Building envelopes to which the BIPV module can be applied are mostly distributed on the building façade; the area of the façade increases as the building height increases. When the BIPV module is applied to the building façade, its performance can be degraded by the influence of installation angle and surrounding shielding materials. Therefore, it is necessary to examine the performance degradation factors of BIPV systems and analyze the characteristics of each performance degradation factor [9–12].

Accordingly, this study proposes a method for applying BIPV systems to non-residential zero energy buildings and will generate basic data for BIPV installation by analyzing the long-term power generation performance and performance degradation factors of BIPV systems in the Carbon Zero Building of the National Institute of Environmental Research (NIER), South Korea, which was designed as a zero energy building.

## 2. Overview of the Carbon Zero Building

Table 1 and Figure 1 show the construction overview and a photograph of the Carbon Zero Building, respectively. The Carbon Zero Building is located at a latitude of 37.57° and a longitude of 126.64°E, and was constructed in Gyeongseo-dong, Incheon, South Korea in April 2011. It has a total floor area of 2449.24 m$^2$ with one underground floor and two above ground floors. It includes laboratories, large meeting rooms, small meeting rooms, data rooms, and male/female lounges used by the building residents, as well as exhibition rooms, international conference rooms, and a lobby for external events and the promotion of zero energy buildings. The cooling and heating area is 1677.94 m$^2$, the area for heating only is 45.26 m$^2$, and the area without cooling and heating is 726.04 m$^2$.

The Carbon Zero Building faces south to minimize the heat loss caused by the envelope and to ensure the largest possible installation of PV modules. On the southern sides and roof of the building, a BIPV module was installed on all sides except for some structural parts and window areas. In addition, the rooftop was designed so that horizontal and inclined PV systems could be installed in the form of a pergola. Moreover, the hot-water supply system and the cooling and heating system in the building uses solar heat and a geothermal heat pump, and all energy used in the Carbon Zero Building is generated by electricity.

**Table 1.** Construction overview of the Carbon Zero Building.

| Floor | Facility | Use | Area (m²) |
|---|---|---|---|
| Underground floor | Public facility | Machine room, electricity room, generator room, control room | 521.6 |
| 1st floor | Promotion facility | Exhibition room | 280.7 |
| | | International conference room | 431.4 |
| | | Information room, situation room | 18.0 |
| | | Warehouse | 11.3 |
| | Public facility | Rest room | 22.6 |
| | | Hall | 138.3 |
| | | EV | 28.3 |
| | | Wind-proof room | |
| 2nd floor | Research facility | Laboratory | 412.1 |
| | | Data center | 42.7 |
| | | Large meeting room | 92.0 |
| | | Small meeting room | 22.8 |
| | | Data storage room | 56.1 |
| | | Lounges (male and female) | 30.4 |
| | Public facility | Rest room | 22.6 |
| | | Hall | 153.6 |
| | | EV | |
| | | Hallway | 164.9 |
| | | Staircase | |
| Total floor area | | | 2449.2 |

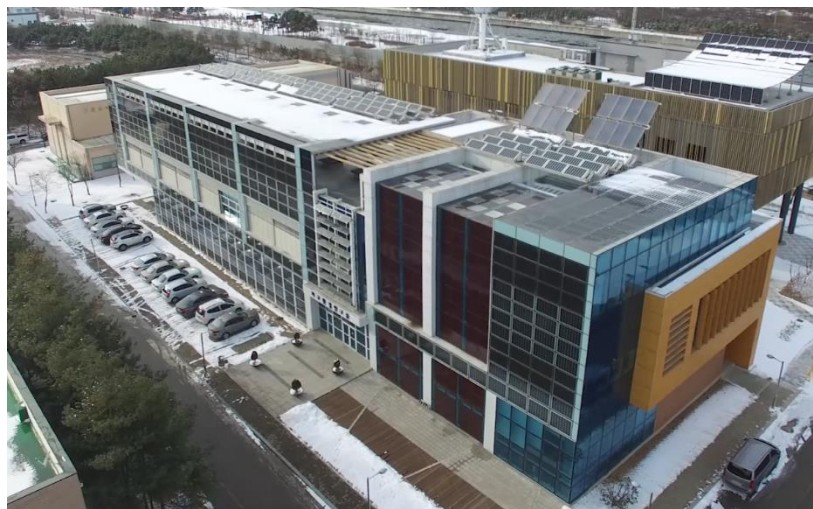

**Figure 1.** Photograph of the Carbon Zero Building.

## 3. BIPV Systems in the Carbon Zero Building

### 3.1. BIPV Systems

The total capacity of the PV system applied to the Carbon Zero Building was 116.241 kWp. The system was composed of individual systems using a total of 15 inverters from 3 kWp to 45 kWp. Four types of polycrystalline modules and one type of amorphous module were used for the PV system. The module types were glass to glass (GtoG), glass to Tedlar/crystal (GtoT), and amorphous. Low-efficiency transparent thin film modules were also applied for lighting in the building. Table 2 shows the specifications of the modules installed in the Carbon Zero Building.

**Table 2.** Performance overview of the installed modules.

| Type | Polycrystal | | | | Amorphous |
|---|---|---|---|---|---|
| Voc | 7.6 | 25.5 | 30.6 | 34.5 | 91.8 |
| Vmp | 5.3 | 17.8 | 21 | 26.1 | 66 |
| Isc | 8.92 | 8.92 | 8.92 | 8.92 | 1.09 |
| Imo | 8.17 | 8.17 | 8.17 | 8.17 | 0.75 |
| Efficiency (%) | 8.32 | 8.14 | 9.77 | 11.65 | 5.3 |
| Capacity (W) | 43 | 145 | 171 | 213 | 50 |
| Number of photovoltaic (PV) Cells | 12 | 40 | 48 | 54 | - |
| Module quantity | 32 | 590 | 37 | 76 | 136 |

In the Carbon Zero Building, a total of 871 modules of five different types were installed. The installation area was 1,177.9 m$^2$. Fifteen arrays were connected according to module type, installation location, and capacity. Table 3 shows the detailed specifications of the installed PV arrays. Their installation locations are shown in Figures 2 and 3, where the labels represent the name of the array connected to each inverter.

**Table 3.** PV module installation overview.

| Inverter Number | Module Type | Tilt Angles | Module Area (m$^2$) | Module Capacity (Wp) | Module Quantity | Total Area (m$^2$) | Total Capacity (kWp) | Inverter Capacity (kWp) |
|---|---|---|---|---|---|---|---|---|
| INV_1 | | 90 | 1.84 | 171 | 12 | 22.08 | 2.052 | 3 |
| INV_2 | | 90 | 1.84 | 171 | 12 | 22.08 | 2.052 | 3 |
| INV_3 | Glass to Glass | 90 | 1.84 | 171 | 13 | 23.92 | 2.223 | 3 |
| INV_4 | | 90 | 1.77 | 213 | 10 | 17.7 | 2.13 | 3 |
| INV_5 | Glass to Glass | 0 | 1.41 | 145 | 18 | 25.38 | 2.61 | 3 |
| INV_6 | | 0 | 1.41 | 145 | 18 | 25.38 | 2.61 | 3 |
| INV_7 | Glass to Tedlar/crystal | 30 | 1.41 | 145 | 16 | 22.56 | 2.32 | 3 |
| INV_8 | | Tracking | 0.53 | 43 | 32 | 16.96 | 1.376 | 3 |
| INV_9 | | 0 | 0.93 | 50 | 32 | 29.76 | 1.6 | 3 |
| INV_10 | Amorphous | 90 | 0.93 | 50 | 52 | 48.36 | 2.6 | 3 |
| INV_11 | | 90 | 0.93 | 50 | 52 | 48.36 | 2.6 | 3 |
| INV_12 | Glass to Tedlar/crystal | 30 | 1.41 | 145 | 70 | 98.7 | 10.15 | 11 |
| INV_13 | | 90 | 1.77 | 213 | 66 | 116.82 | 14.058 | 15 |
| INV_14 | Glass to Glass | 90 | 1.41 | 145 | 160 | 225.6 | 23.2 | 25 |
| INV_15 | | 0 | 1.41 | 145 | 308 | 434.28 | 44.66 | 45 |
| Total | - | - | - | - | 871 | 1177.94 | 116.241 | 129 |

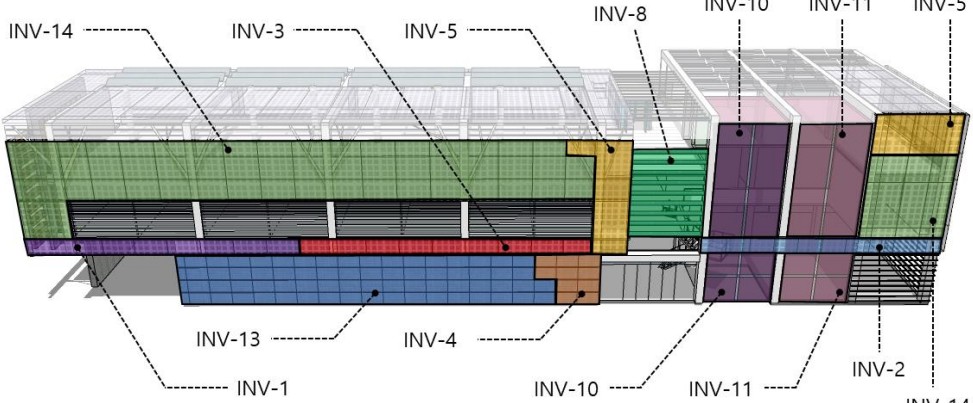

**Figure 2.** Detailed diagram of the building-integrated photovoltaic (BIPV) array installation on the southern façade of the Carbon Zero Building.

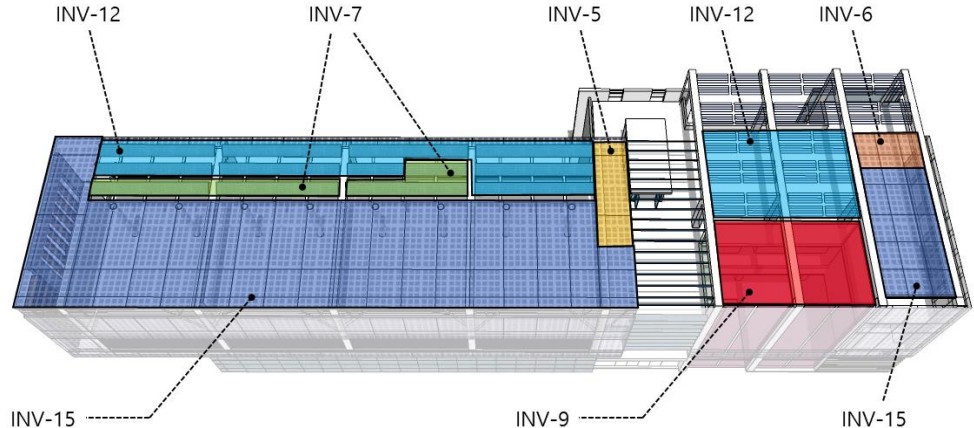

**Figure 3.** Detailed diagram of the BIPV array installation on the roof of the Carbon Zero Building.

*3.2. Monitoring System*

Figure 4 shows the electric power system and monitoring system of the Carbon Zero Building. To investigate the power generation characteristics of each array, data were collected by connecting a power monitoring module bus to 15 PV inverters. To measure the total generated power, an additional watt-hour meter was installed at the front of the transformer where the power generated from each inverter was collected. Moreover, data required for the detailed analysis of the BIPV systems were secured by installing pyranometers, thermometers, and hygrometers. In addition, a watt-hour meter was installed to measure the electric power used in the building and the power transmitted to and received from the power system. These sensors were connected to the integrated management system to collect data in real time.

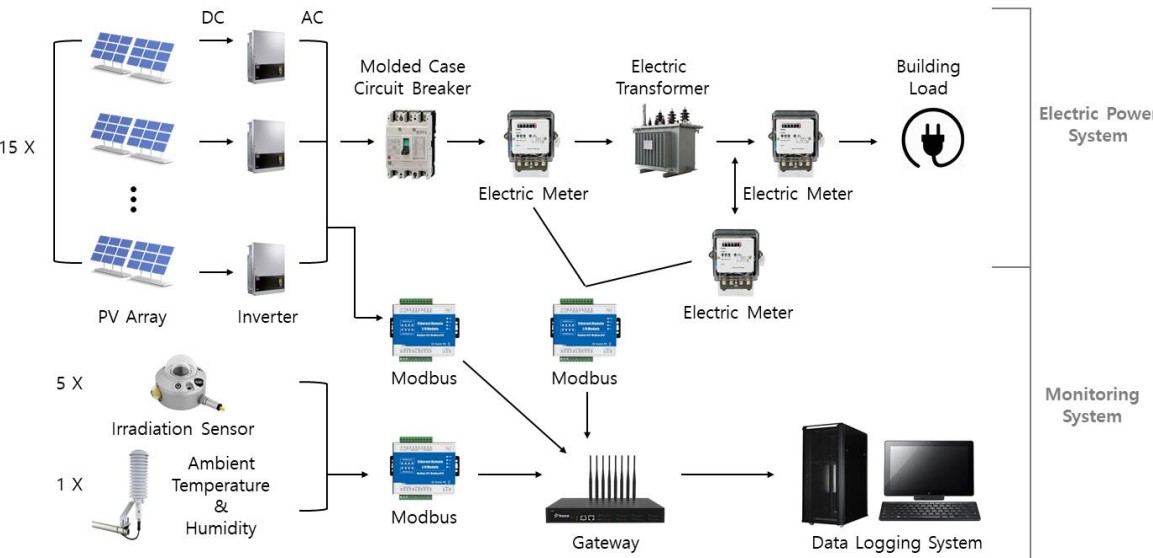

**Figure 4.** Flow diagram of the electric power system and monitoring system of the Carbon Zero Building.

## 4. BIPV System Performance Analysis

*4.1. Data Analysis Period*

The data analysis period was five years from 1 January 2013 to 31 December 2017. Data from 24 days when the PV system and monitoring system were cut off for power safety inspection and system inspection were excluded. However, the total generated power was determined even during the period when the monitoring program was not operational because the accumulated power value was collected by each inverter and watt-hour meter. The data collection period was 1826 days, the

number of daily data files for the 1802 days (excluding the 24 non-operational days) was 28,832, and the total number of data points was 41,518,080. For a quantitative analysis of the PV system, the required data were extracted and analyzed using the Microsoft Visual Basic for Application program.

*4.2. Data Analysis Method*

Analysis of the BIPV systems was conducted using the IEC standard 61724 calculation method [13]. The capture loss ($L_C$) was utilized as a performance evaluation index to analyze the difference between the energy production under the standard test conditions (STC) and the actual energy production. It was used to express the amount of loss due to mismatching, partial shading, module temperature, cable loss, and installation angle of the maximum power point tracking (MPPT) control. It is expressed in Equation (1).

$$L_C = Y_R - Y_A \tag{1}$$

where $Y_A$ represents the energy output (DC) value generated per PV array in kW, expressed in Equation (2), and $Y_R$ represents the reference yield, which can be expressed as the ratio of the total in-plane irradiation per day to the reference in-plane irradiance, as shown in Equation (3).

$$Y_A = \frac{E_{DC}}{P_o} \tag{2}$$

$$Y_R = \frac{H_{Id}}{G_I} \tag{3}$$

Moreover, the system loss ($L_S$) is an important evaluation element used for analyzing the efficiency degradation that occurs when the DC power produced by each module is converted into AC power by the installed inverters. It is expressed in Equation (4).

$$L_S = Y_A - Y_F \tag{4}$$

where $Y_F$, which is the final yield in Equation (4), represents the energy output (AC) generated per installed PV array in kW as shown in Equation (5).

$$Y_F = \frac{E_{AC}}{P_o} \tag{5}$$

The performance ratio (PR) represents the ratio of the actual power generation performance to the ideal power generation performance that does not consider losses under STC, as shown in Equation (6). Power generation performance degradation occurs due to various factors, such as the inverter's loss, the effect of partial shading, and mismatching of the operating PV system. These factors are analysis items that can evaluate the power generation performance degradation of the PV system in a relatively simple and accurate manner [7,8].

$$P_R = \frac{Y_F}{Y_R} \tag{6}$$

Here, $N_t$, which is the efficiency of the module, represents the ratio of the actual energy production (AC) to the solar radiation irradiated onto the total area of the PV module, as shown in Equation (7).

$$N_t = \frac{E_{AC}}{H_{Id} \times A} \tag{7}$$

## 5. BIPV System Performance Results

*5.1. Total Energy Production of the BIPV Systems*

Figure 5 shows the monthly energy production of the BIPV systems in the Carbon Zero Building from January 2013 to December 2017. The cumulative generated in Figure 5 is given on the secondary

axis on the right, whereas the monthly cumulative generated is given on the primary axis on the left. The monthly cumulative generated on the primary axis is represented by a line plot and compares the monthly cumulative generated. The bar plot denotes the cumulative energy generated given on the secondary axis and illustrates the annual trend of monthly cumulative generated. The five-year average monthly energy production of the BIPV systems was 8288 kWh, indicating that 497,262 kWh of power was produced over the five years. The lowest power (94,808 kWh) was produced in 2016 and the highest power (103,732 kWh) in 2013. The difference between the highest and lowest energy production was 8925 kWh, representing 8.97% of the average annual energy production of 99,452 kWh. The average monthly energy production during the analysis period was 8288 kWh, although January, February, July, November, and December exhibited lower energy production than the average. This is because solar radiation is low in winter and high in spring and autumn in South Korea. In particular, energy production in July did not reach the average because July is the rainy season in South Korea. Energy production in November and December was significantly lower in 2015 than in other years. This is because the BIPV systems in the Carbon Zero Building were not operational for approximately 10 days from 24 November to 5 December 2016 during an electric safety diagnosis of the building.

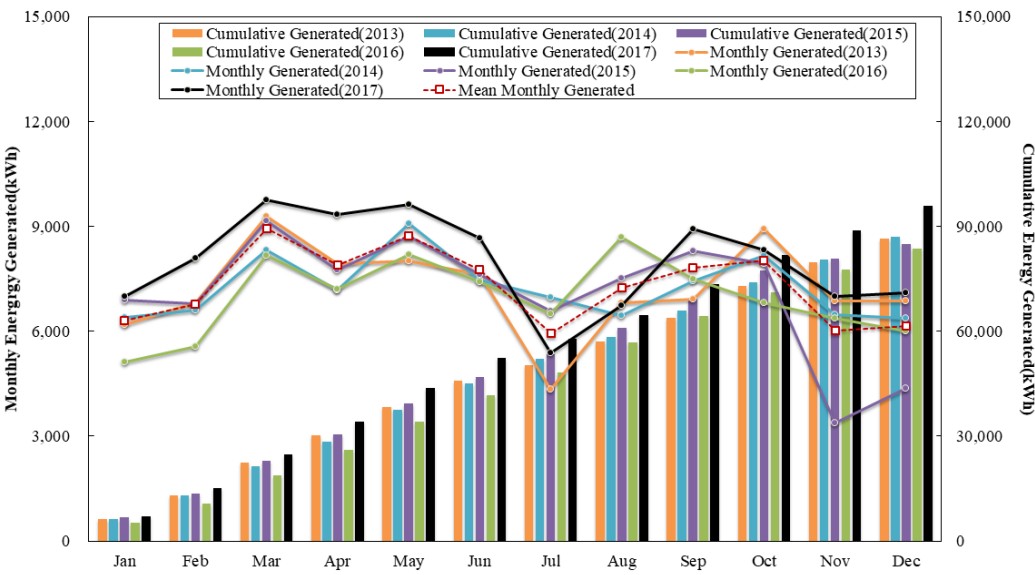

**Figure 5.** Monthly and accumulated energy production.

## 5.2. Comparison between Planned and Actual Energy Production of the BIPV Systems

Figure 6 shows the measured energy production compared to the predicted energy production over the five-year period. The bar plot in this figure represents the monthly energy generated scaled on the primary axis on the left, and it shows the difference between simulated and measured data. The cumulative energy generated denoted by a line plot has been included to quantify the difference between simulated and monitored data over the course of five years. In the building planning stage of the Carbon Zero Building, the energy production of the BIPV systems was predicted through simulations before introducing the system. To predict the energy production of the BIPV systems, the simulation was performed using the 30-year average data of Incheon, where the Carbon Zero Building was located, published by the Korea Meteorological Administration (KMA) of South Korea. As a result, the annual average energy production was predicted to be 975.2 kWh/kWp. The measured energy production of the BIPV systems over the five-year-period, however, was 892.4 kWh/kWp in 2013, 883.7 kWh/kWp in 2014, 858.2 kWh/kWp in 2015, 815.6 kWh/kWp in 2016, and 828.0 kWh/kWp in 2017. The five-year average energy production was 855.6 kWh/kWp, which was 119.6 kWh/kWp lower than the predicted energy production. The predicted five-year energy production was 566,778 kWh, but the measured five-year energy production was 497,262 kWh, which was approximately 12.26%

lower, despite the prediction using previous solar radiation and outdoor temperature data provided by the KMA. As this was due to efficiency degradation during actual application of the BIPV systems, it is necessary to identify the cause of this efficiency degradation through a detailed analysis.

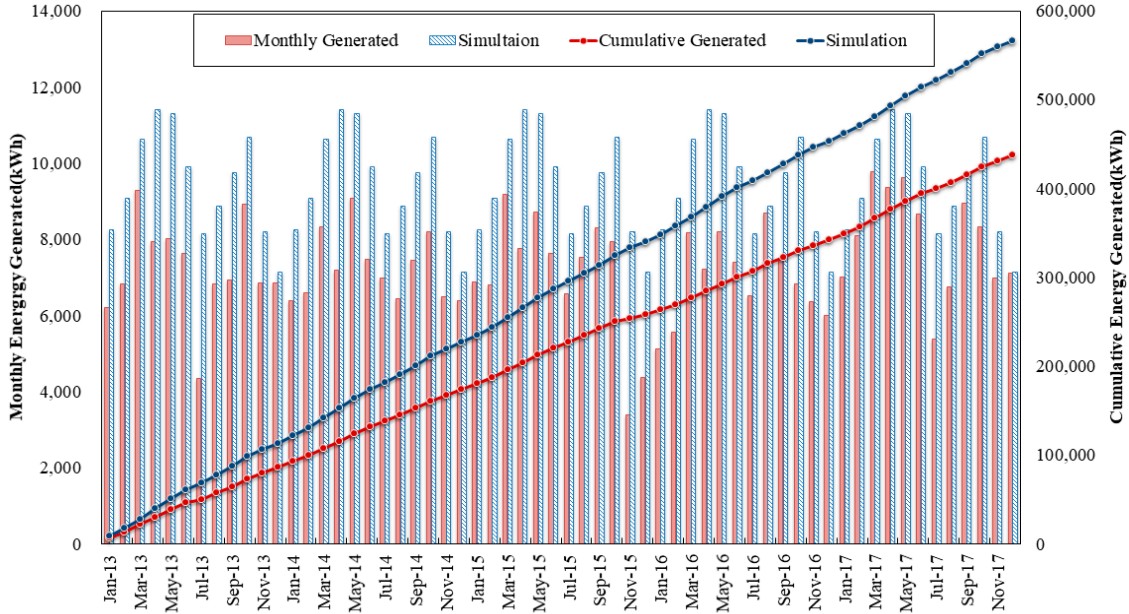

**Figure 6.** Comparison of measured and simulated five-year and monthly energy production values.

### 5.3. Yields and Losses of the BIPV Systems

Figure 7 shows the monthly final yields, capture losses, and system losses of all BIPV systems installed in the Carbon Zero Building from January 2013 to December 2017. The system loss ($L_C$) represents the time of power generation loss during conversion of the DC power produced by each module into AC power by the installed inverters. The capture loss ($L_S$) analyzes the difference between the energy production under STC and the actual energy production. It represents the time of power generation loss during power production from the modules to the inverters for the mismatching, partial shading, module temperature, cable loss, and installation angle of the MPPT control [14,15]. The five-year average final yield of the BIPV systems installed in the Carbon Zero Building was 2.00 h/d, and the capture loss was 1.50 h/d. The system loss was found to be 0.21 h/d. Figure 8 shows the capture losses, system losses, and final yields of the PV systems classified by inverter number for five years from 2013 to 2017. The system losses by inverter ranged from 0.14 to 0.31 h/d. There was no significant difference between inverters regarding the conversion performance of produced DC power to AC power. In the case of the capture loss, however, significant differences occurred from 0.21 to 1.81 h/d. This is because the installed BIPV systems were significantly affected by the installation inclination angle and PV type. The capture loss was 1 h/d or less for the crystal system modules, which were installed with inclination angles of 30° and 90°. The arrays with an installation angle of 0°, tracking mode arrays, and thin-film transparent modules exhibited a capture loss of 1 h/d or greater. Moreover, Inv-10 in Figure 8 is the BIPV thin-film transparent system installed on the wall of the Carbon Zero Building. The system was damaged before January 2013, and the power generation system has since been non-operational. In the case of the tracking array, the capture loss was higher than that of the 30° fixed-type array because the tracking system also failed before 2013 and the array inclination angle has since been maintained at 10°. The total value presented in Figure 8 is the five-year average value of the entire system, including the systems that failed, and is higher than the capture loss values of general PV systems. Figures 9–14 shows the STC efficiency, system efficiency, and PR characteristics and average of the photovoltaic power generation system for each year in detail.

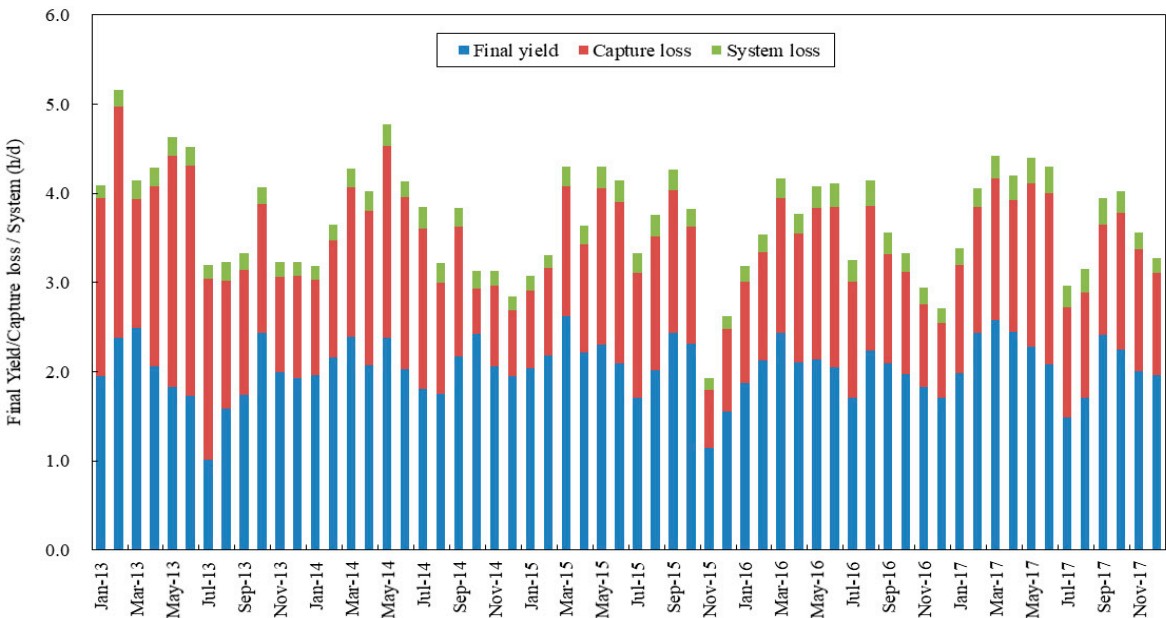

**Figure 7.** Average monthly final yields, capture losses, and system losses of the entire BIPV system.

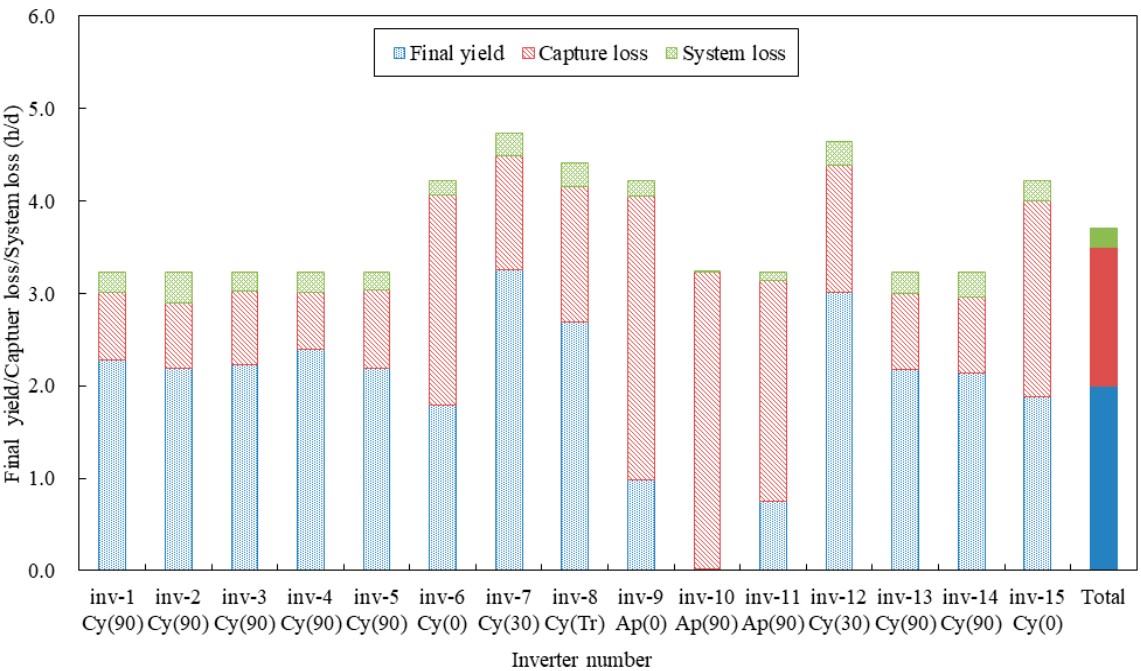

**Figure 8.** Five-year average final yields, capture losses, and system losses for different inverters.

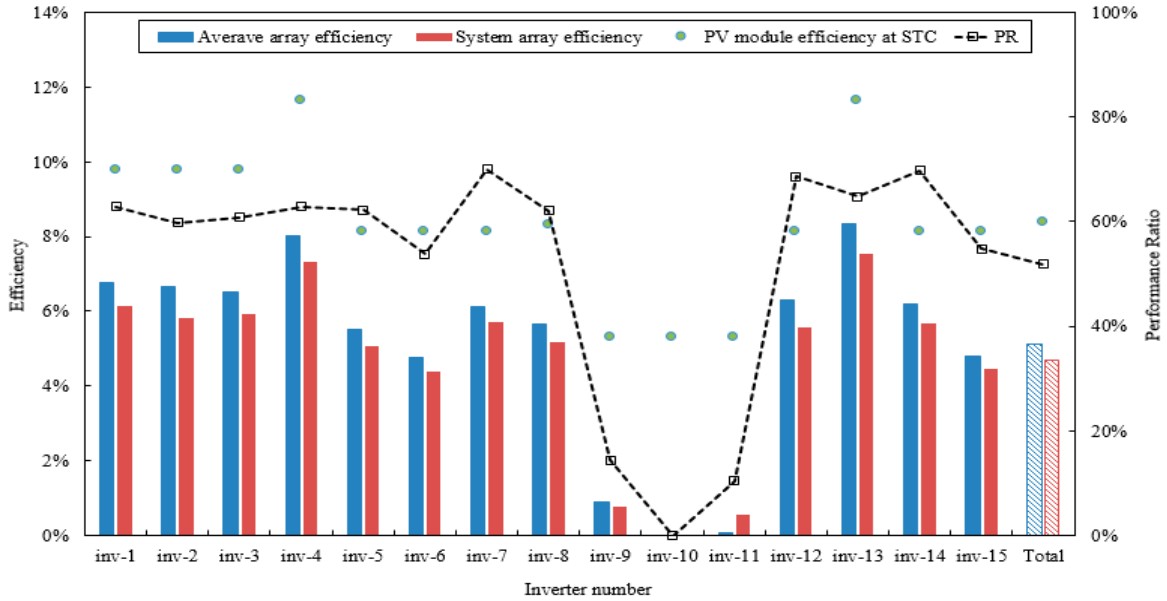

**Figure 9.** PV power generation efficiency and PR by 2013.

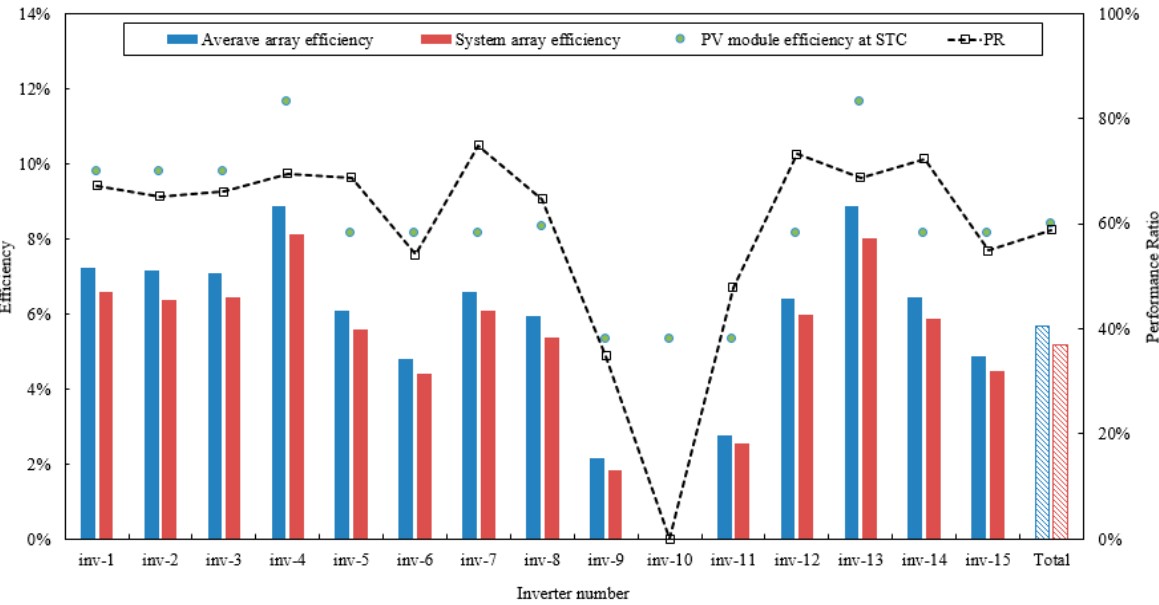

**Figure 10.** PV power generation efficiency and PR by 2014.

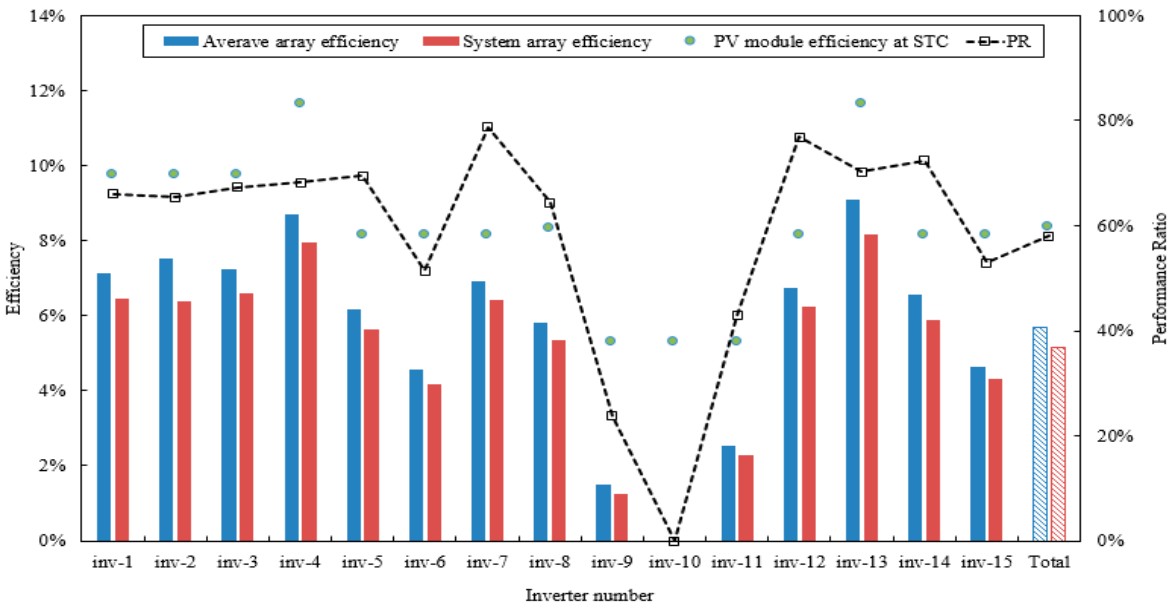

**Figure 11.** PV power generation efficiency and PR by 2015.

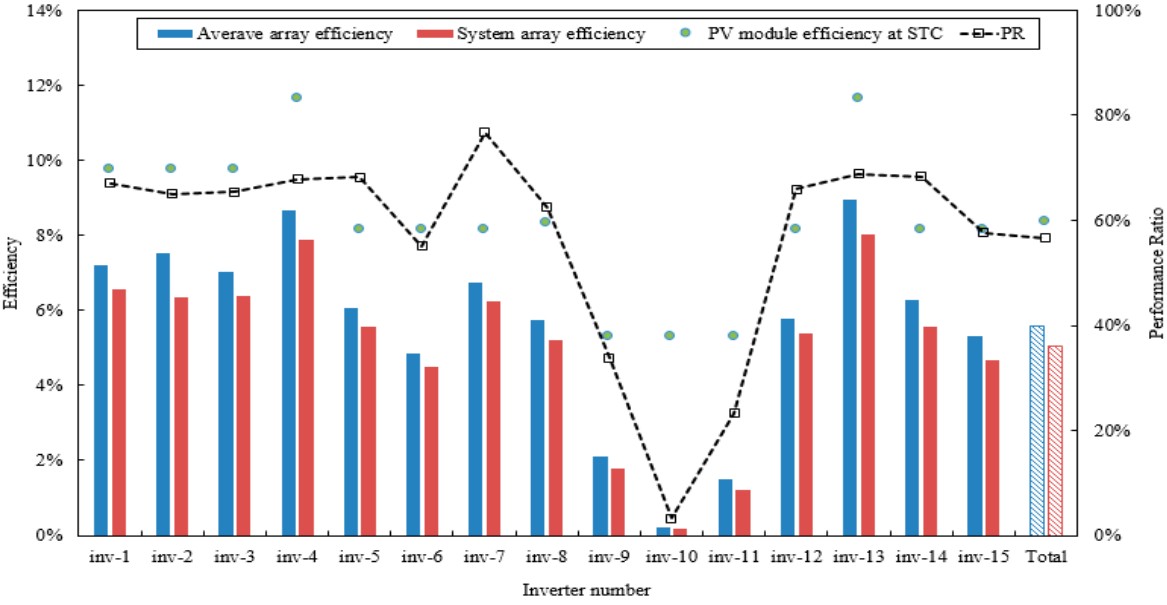

**Figure 12.** PV power generation efficiency and PR by 2016.

*5.4. Efficiency Degradation of the BIPV Systems in the Carbon Zero Building*

The system losses ranged from 0.14 to 0.31 h/d; i.e., there was no significant influence on the inverters that converted DC into AC. This indicates that the efficiency degradation of the BIPV systems in the Carbon Zero Building was mostly caused by capture losses. The capture losses are power generation losses that occur in the power generation process from the modules to the inverters and are caused by many variables. Therefore, the causes of power generation degradation were analyzed for the horizontally installed arrays (Inv-6, Inv-9, Inv-15) and vertically installed thin-film arrays (Inv-10, Inv-11) with relatively high capture losses.

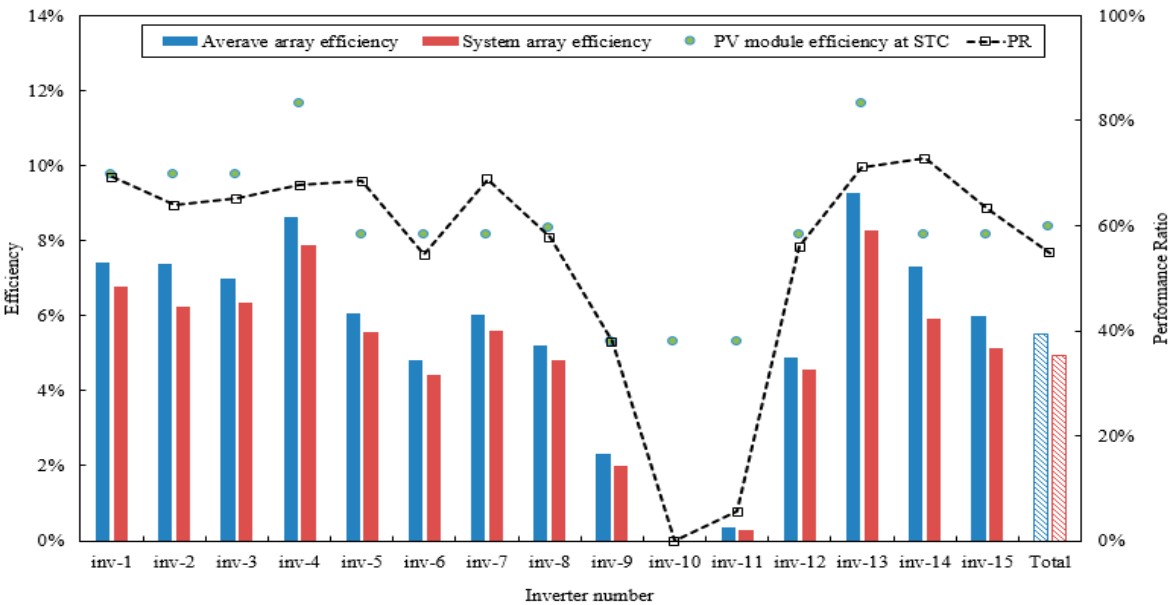

**Figure 13.** PV power generation efficiency and PR by 2017.

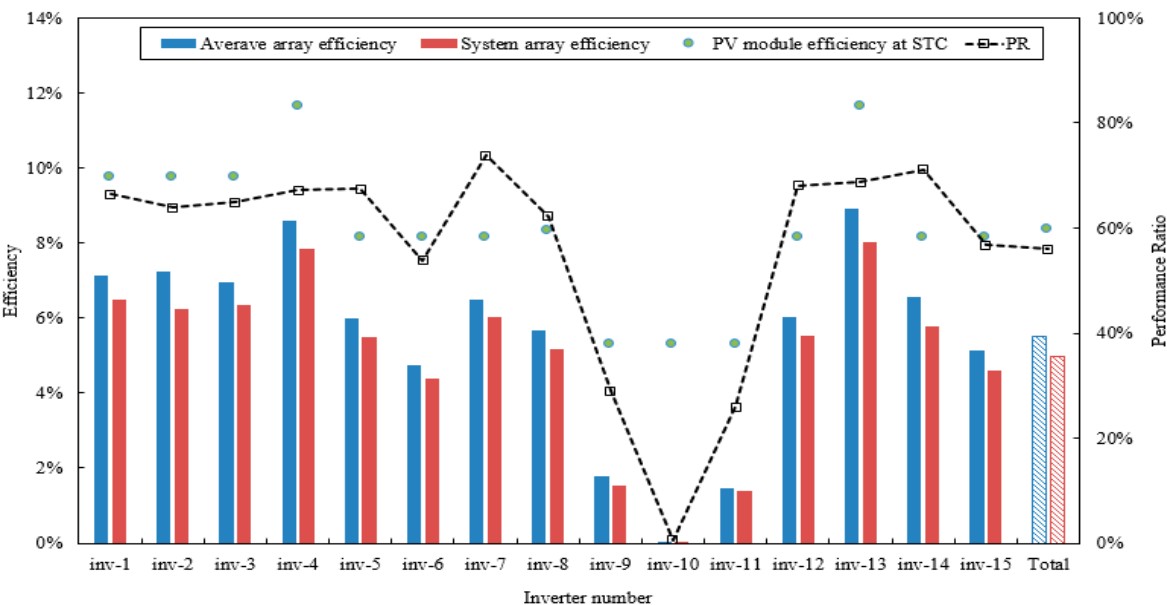

**Figure 14.** PV power generation efficiency and PR by 5 year average (2013–2017).

5.4.1. Efficiency Degradation of Horizontal BIPV Modules due to Dust Accumulation

In Figure 15, the daily precipitation and efficiency variations of the Inv-15 array, which was horizontally installed with a capacity of 44.66 kWp, is analyzed to identify the causes of efficiency degradation of the horizontal PV modules due to dust. The analysis was conducted for two months from April to May in 2017 when solar radiation was high and precipitation was intermittent.

From April to May in 2017, the number of rainy days in Incheon was 15. Among them, six days had daily precipitation of 4 mm or more. The daily power generation efficiency was analyzed before and after these high rainfall days and revealed that rainfall increased the power generation efficiency by an average of more than 9.5%. In particular, on April 7, 2017 when precipitation of more than 4 mm occurred for two consecutive days, the system power generation efficiency increased by more than 15% compared with April 4. Thus, changes in daily power generation efficiency were directly affected by precipitation. In the case of the horizontal modules installed in the Carbon Zero Building, the power

generation efficiency was degraded as dust accumulated (Figure 16), but the precipitation removed the dust accumulated on the front of the PV modules and increased the power generation efficiency.

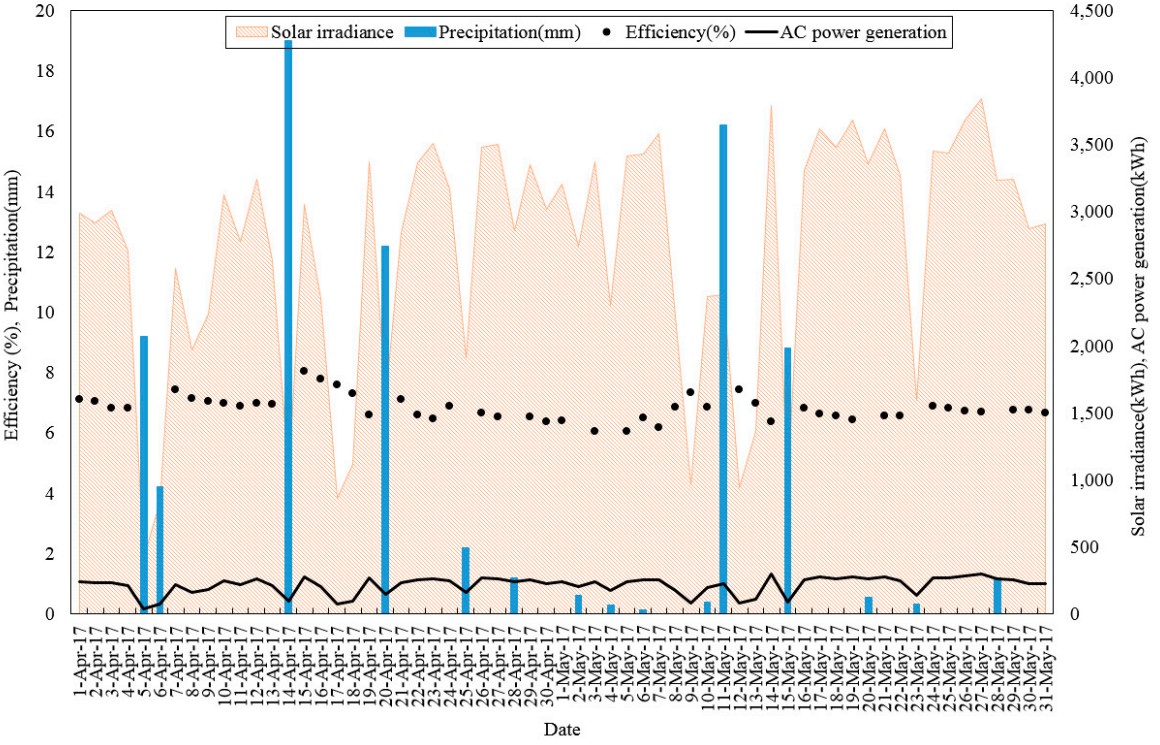

**Figure 15.** Daily efficiency and daily precipitation of the Inv-15 array.

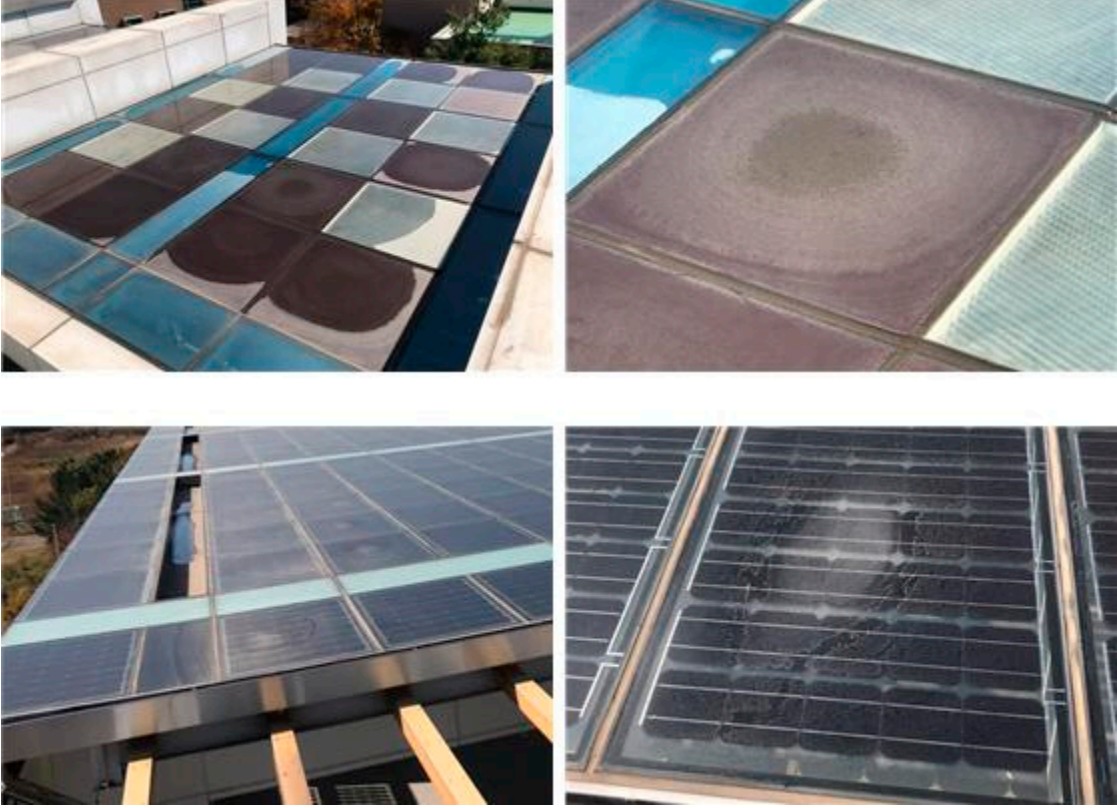

**Figure 16.** Dust accumulation on the front of the horizontal BIPV modules in the Carbon Zero Building (Top left, Inv-9; top right, Inv-9 details; bottom left, Inv-15; bottom right, Inv-15 details).

### 5.4.2. Causes of Output Degradation of the Thin-Film Modules

The amorphous BIPV module installed in the Carbon Zero Building consisted of 2.6 kW Inv-10 and Inv-11 systems installed on the southern side at 90° and a horizontally installed 1.6 kW Inv-9 system. Among the vertically installed amorphous BIPV systems, Inv-10 exhibited five-year energy production of zero due to the system failure that occurred before the period of this study. Figure 17 compares the energy production per kWp between Inv-9 (horizontally installed amorphous BIPV system) and Inv-14 (horizontally installed GtoG BIPV system) for seven days from 6 March to 12 March in 2017. Figure 18 compares the energy production per kWp between Inv-11 (vertically installed amorphous BIPV system) and Inv-15 (vertically installed GtoG BIPV system) during the same period.

As shown in Figures 17 and 18, the energy production of Inv-14 and Inv-15, which were GtoG BIPV systems, exhibited a constant output in proportion to the solar radiation, whereas that of Inv-9 and Inv-11, which were amorphous BIPV systems, did not show constant output, and the energy production was significantly lower [16,17]. Unlike crystal system modules with MPPT-controlled crystal system solar cell inverters that consider only direct solar radiation, amorphous modules simultaneously react to direct and scattered solar radiation; therefore, MPPT-controlled inverters for amorphous modules must consider both direct and scattered solar radiation [18–21]. As the same inverters used for the crystal system modules were applied to the amorphous modules in the Carbon Zero Building, the energy production was significantly lower because the control characteristics of the amorphous modules were not considered.

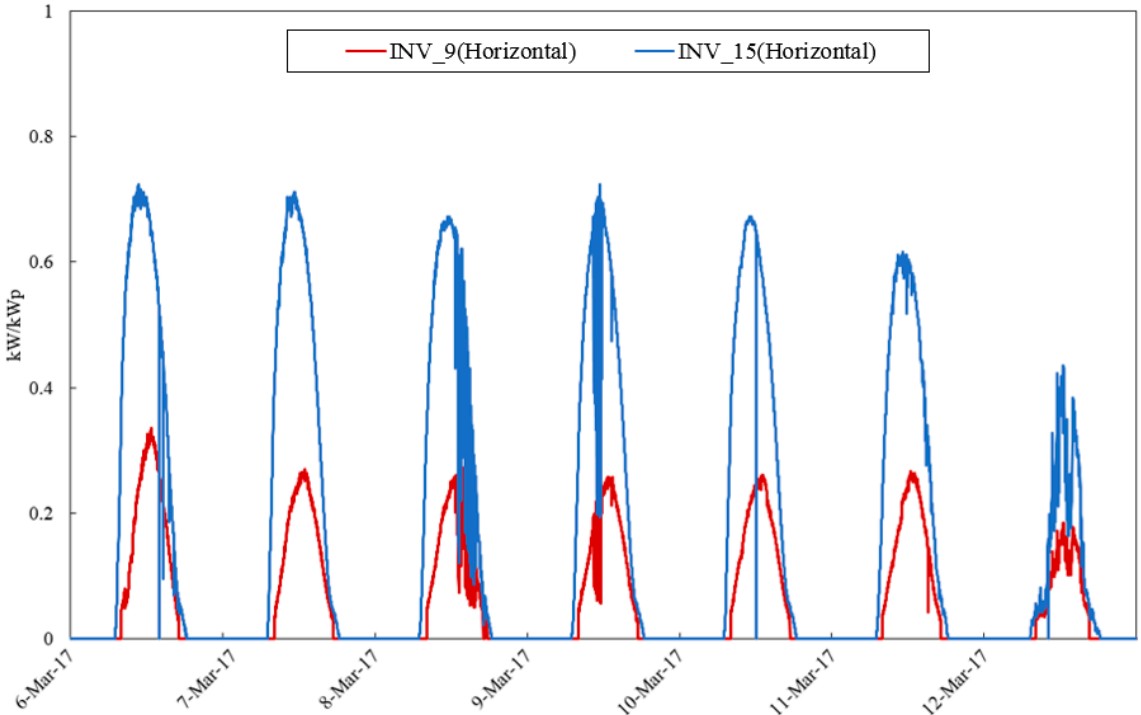

**Figure 17.** Comparison of energy production per kWp between horizontal BIPV systems (Inv-9 and Inv-15).

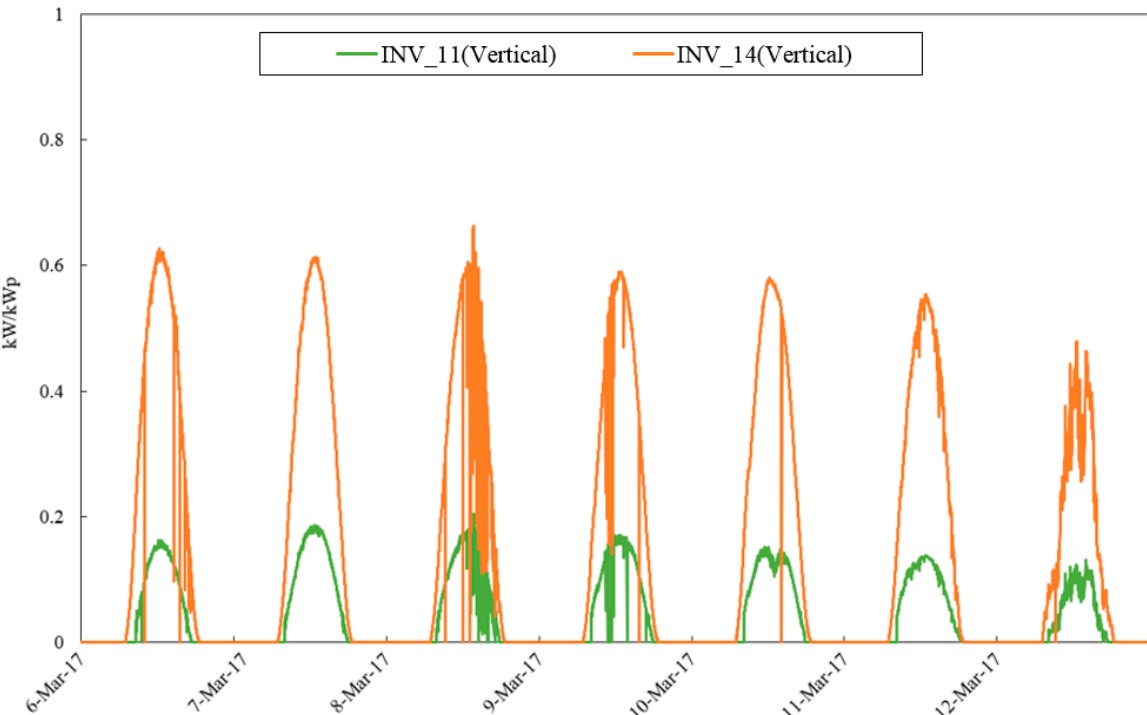

**Figure 18.** Comparison of energy production per kWp between vertical BIPV systems (Inv-11, Inv-14).

## 5.5. Energy Production by BIPV Installation Angle

Figure 19 shows the energy production per kWp as a function of different installation angles of the same BIPV systems in the Carbon Zero Building from January 2013 to December 2017. The monitored data for Inv-12 (30°), Inv-14 (90°), and Inv-15 (0°) with identical 145Wp-capacity PV modules were converted into kWp for comparison purposes. The BIPV systems with an inclination angle of 30° exhibited the highest energy production under the same solar irradiation, whereas vertical BIPV systems showed the lowest energy production under the same solar irradiation. The correlation between energy production per kWp and solar radiation was $y = 0.0011x − 0.0026$ for the BIPV systems with an inclination angle of 30°, $y = 0.0008x − 0.0018$ for horizontal BIPV systems, and $y = 0.0007x − 0.0017$ for vertical BIPV systems. Therefore, the BIPV systems with an inclination angle of 30° generated approximately 57% more power than vertically (90°) installed systems under the same solar radiation. Moreover, the energy production of horizontal BIPV systems (0°) was up to 14% higher than that of vertical systems.

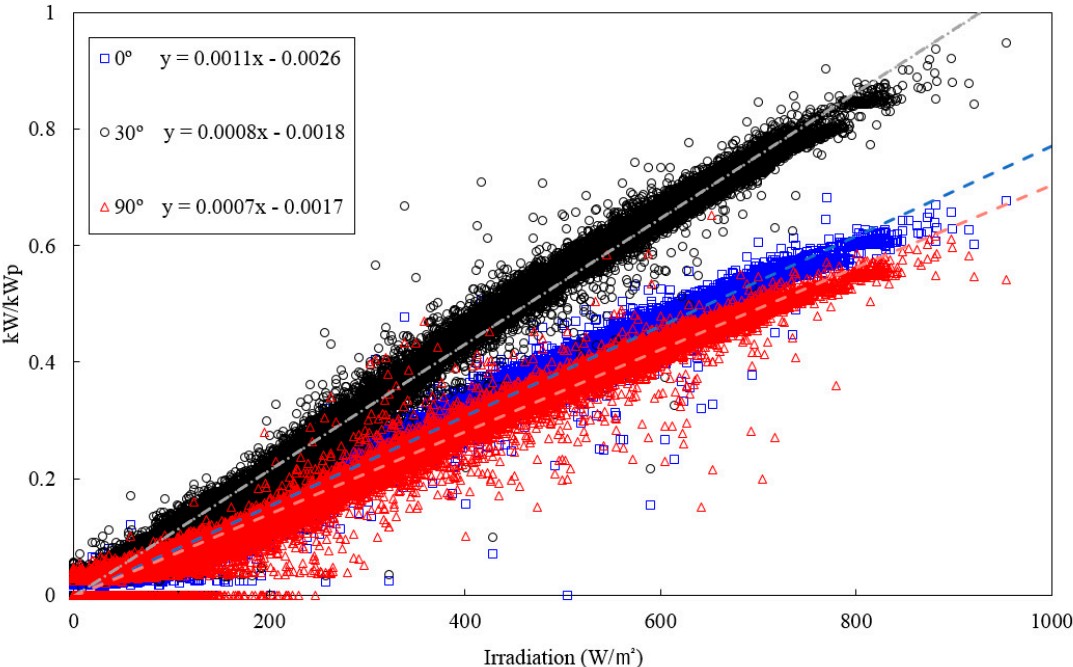

**Figure 19.** Comparison of energy production per kWp for different installation angles of BIPV systems in the Carbon Zero Building.

## 6. Conclusions

In this study, the long-term operation performance of building-integrated photovoltaic (BIPV) systems installed in the Carbon Zero Building was analyzed from 1 January 2013 to 31 December 2017. The following conclusions were drawn.

(1)  Based on simulations, the energy production of the BIPV systems in the Carbon Zero Building was predicted to be 975.2 kWh/kWp. The monitoring results for the five-year period, however, showed that the average annual energy production per kWp was 855.6 kWh/kWp, which was 12.26% lower than the predicted value.

(2)  After analyzing the causes of degradation in the measured energy production, it was concluded that the system losses by inverter, which ranged from 0.14 to 0.31 h/d, indicated no significant difference between inverters. Conversely, the capture losses ranged from 0.21 to 1.81 h/d, indicating significant differences.

(3)  The primary cause of the increase in capture losses was a temporary decrease in the power generation efficiency of the horizontally installed modules due to dust accumulation on the front of the PV modules, which was subsequently removed by precipitation.

(4)  The second cause of the increase in capture losses was the fact that the inverters installed in the Carbon Zero Building could not consider the control characteristics of the amorphous modules, leading to a significant decrease in energy production because the same inverters were used for both crystal system modules and amorphous modules.

(5)  Therefore, the BIPV systems with an inclination angle of 30° exhibited approximately 57% higher energy production than the vertically (90°) installed systems under the same solar radiation. Furthermore, the horizontal (0°) BIPV systems exhibited up to 14% higher energy production than the vertical BIPV systems.

These results indicate that, unlike general PV systems, BIPV power generation performance is affected by various causes. It is expected that the results of this study can be used for the installation and efficiency improvement of future BIPV systems.

**Author Contributions:** Conceptualization, W.J.C., H.J.J., J.-W.P., S.-k.K., and J.-B.L.; methodology, W.J.C., H.J.J., J.-W.P., S.-k.K., and J.-B.L.; validation, W.J.C., H.J.J., J.-W.P., S.-k.K., and J.-B.L.; formal analysis, W.J.C., H.J.J., J.-W.P., S.-k.K., and J.-B.L.; data curation, W.J.C., H.J.J., J.-W.P., S.-k.K., and J.-B.L.; writing—original draft preparation, W.J.C., H.J.J., J.-W.P., S.-k.K., and J.-B.L.; writing—review and editing, W.J.C., H.J.J., J.-W.P., S.-k.K., and J.-B.L.; visualization, W.J.C., H.J.J., J.-W.P., S.-k.K., and J.-B.L. All authors have contributed equally on this manuscript together and all authors have read and approved the final manuscript.

**Funding:** This work was supported by a grant from the National Institute of Environmental Research (NIER) and was funded by the Ministry of Environment (MOE) of the Republic of Korea (NIER-2017-01-02-038).

**Conflicts of Interest:** The authors declare no conflict of interest.

## List of Symbols

| | |
|---|---|
| YA | Array yield [h/d] |
| YF | Final yield [h/d] |
| YR | Reference yield [h/d] |
| EDC | Array DC energy output per year [kWh/year] |
| EAC | AC energy output to the grid per year [kWh/year] |
| GI | Reference irradiation (1000 W/m$^2$) |
| HId | Total in-plane irradiation per year [kWh/m$^2$/year] |
| LC | Capture loss [h/d] |
| LS | System loss [h/d] |
| RR | Performance ratio [%] |
| NE | Reference array efficiency [%] |
| A | Area of array [m$^2$] |
| Ntotal | Overall system efficiency [%] |
| TFE | Total facility energy use [kWh/month] |
| FEL | Facility total energy load met by photovoltaic (PV) production [kWh/month] |
| Po | Installed capacity [kWp] |

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
