# Peer review of "Power Generation Performance of Building-Integrated Photovoltaic Systems in a Zero Energy Building"

_energies, doi:10.3390/en12132471_

Round 1
Reviewer 1 Report
Check and improve English (correct for example spelling for “Simultaion”). What is kWp ? Try to keep the entire table on the same page (Table 3 is split on two pages). Also the Figure title should be on the same page with the figure. Explain what is P0 from relations (2) and (5). Figure 5 needs more detailed explanation. It is not clear for example why the cumulative generated for January 2013 is 500kWh and the monthly generated for January 2013 is over 6000KWh? What represents the monthly generated and the monthly cumulative generated (which is smaller)? The same comments for Figure 6 monthly generated for January is over 6000 KWh and the cumulative is around 100 Kwh, explain what is cumulative generated, and why is so small in comparison with the monthly generated. Quality of Figure 9 must be improved. In Figure 10, the labels on X axis are very dense becoming unclear. Please fill the empty space from each page with valuable comments.Author Response
Thank you for your valuable feedback.
Yes, the manuscript has been proofread and revised by a native English speaker.
The generation capacity of a PV module is determined by the power output measured under the STC (Standard Test Condition) as a function of solar radiation. This power output is given in kWp.
Yes, the table and the figure have been resized to fit in one page as you suggested.
Yes, the description of Po is given under List of symbols.
In Figure 5, the cumulative generated in Figure 5 is given on the secondary axis on the right, whereas the monthly cumulative generated is given on the primary axis on the left. The monthly cumulative generated on the primary axis is represented by a line plot and compares the monthly cumulative generated. The bar plot denotes the cumulative energy generated given on the secondary axis and illustrates the annual trend of monthly cumulative generated. These sentences have been added to the manuscript.
The bar plot in the Figure 6 represents the monthly energy generated scaled on the primary axis on the left and it shows the difference between simulated and measured data. The cumulative energy generated denoted by a line plot has been included to quantify the difference between simulated and monitored data over the course of five years.
Figure 9 and Figure 10 will be improved for clarity.

Reviewer 2 Report
The paper entitled "Power generation performance of building-integrated photovoltaic systems in a zero energy building" provide valuable information. The paper is well written and accepted in present form. Few minor issues should be include before publication.
What PV module (what technology? Silicon?) used in their study should be mentioned in the article.
Improve the Fig.9 qualities
Level each figure in Fig.11 and mention in the text
Fig.12 complete the level (i.e., end the data - 12-March-17 to?). Similar in Fig.13
Author Response
Thank you for your valuable feedback.
The PV module considered in this work was elaborated in Section 3.1 BIPV system as G to G, G to T, Amorphous-type BIPV while its characteristics and schematic are given in Table 3, Figure 2, and Figure 3.
Figure 9 will be improved for clarity.
Additional description of the figure will be given.
The labels in Figure 12 and Figure 13 will be edited up to 13-Mar-17.
Reviewer 3 Report
The authors analyzed five-year performance of BIPV systems located at Carbon Zero Building of the National Institute of Environmental Research of South Korea. The data shown in this manuscript is complete, and presentation is clear. Module performance and losses were compared between different types of systems. The effects of rainfall on PV system performance was also evaluated. I suggest publishing it in Energies after addressing the following two minor flaws:
1) Please define P0 in equation (2) and (5);
2) In section 5.5, or Figure 14, the authors compared the energy production of different installation angles of BIPV systems. However, the authors failed to explain if the systems have same module types (GtoG, GtoT, amorphous). Please clarify.
Author Response
Thank you for your valuable comments.
The description of Po is given under List of symbols.
A sentence elaborating that Figure 14 compares the energy production of a BIPV system for installation angles of INV-12(30°), INV-14(90°), and INV-15 (0°) has been added under Section 5.5.